



# How much solar wind data are sufficient for accurate fluxgate magnetometer offset determinations?

Ferdinand Plaschke[1]

[1]Space Research Institute, Austrian Academy of Sciences, Graz, Austria.

**Correspondence:** Ferdinand Plaschke (ferdinand.plaschke@oeaw.ac.at)

**Abstract.** Accurate magnetic field measurements by fluxgate magnetometers on-board spacecraft require ground and regular in-flight calibrations activities. Therewith, the parameters of a coupling matrix and an offset vector are adjusted; they are needed to transform raw magnetometer outputs into calibrated magnetic field measurements. The components of the offset vector are typically determined by analyzing Alfvénic fluctuations in the solar wind, if solar wind measurements are available. These are

characterized by changes in the field components, while the magnetic field modulus stays constant. In this paper, the following question is answered: How much solar wind data are sufficient for accurate fluxgate magnetometer offset determinations? It is found that approximately $50$ hours of solar wind data are sufficient to achieve offset accuracies of $0.2\,\mathrm{nT}$, and about $20$ hours suffice for accuracies of $0.3\,\mathrm{nT}$ or better, if the magnetometer offsets do not drift within these time intervals and if the spacecraft fields do not vary at the sensor position. Offset determinations with uncertainties lower than $0.1\,\mathrm{nT}$, however, would

require at least hundreds of hours of solar wind data.

## 1 Introduction

In-situ investigations of the plasma environments of planets, moons, comets, or other solar system bodies require magnetic field measurements by spacecraft magnetometers. Typically, fluxgate magnetometers are used for scientific applications. The required measurements can only be provided if those magnetometers are accurately calibrated. This means that a coupling

matrix $\mathbf{C}$ and an offset vector $O$ have to be accurately known in order to transform raw magnetometer outputs $B_{\mathrm{raw}}$ into calibrated magnetic field measurements $B$ (e.g., Fornaçon et al., 1999; Balogh et al., 2001; Auster et al., 2008):

$$B = \mathbf{C} \cdot B_{\mathrm{raw}} - O \tag{1}$$

Both, $\mathbf{C}$ and $O$ should be determined on ground and in-flight, as calibration parameters, in particular the offset components, are known to change over time. Offset changes may be associated with instrument drifts or with variations of the spacecraft-

generated magnetic fields at the magnetometer sensor, as the offsets are the outputs of a magnetometer in vanishing ambient field conditions.

     If the spacecraft is spin-stabilized, then the spin plane offset components are easily determined by minimizing the spin tone content in the despun spin plane magnetic field measurements (e.g. Farrell et al., 1995; Kepko et al., 1996). If the spacecraft is non-spinning, i.e., three axis stabilized, then the following methods can be used for offset determination: (1) Alfvénic



fluctuations that are abundant in the solar wind are characterized by changes in the magnetic field components, while the field magnitude stays constant. Analysis of such fluctuations allows for an adjustment of the offsets by minimization of the changes in the magnetic field magnitude (e.g., Belcher, 1973; Hedgecock, 1975; Leinweber et al., 2008). This is the typical method for offset determination, if solar wind measurements are available. (2) Compressional fluctuations can also be used to determine

magnetometer offsets, by application of the mirror mode method (Plaschke and Narita, 2016; Plaschke et al., 2017). In this case the fact is used that the maximum variance direction of the fluctuations should coincide with the average magnetic field direction. Any mismatch may be attributed to incorrect offsets. (3) Furthermore, offsets may be obtained by comparing fluxgate magnetometer magnetic field measurements to: (i) measurements from an electron drift instrument (EDI) or from an absolute magnetometer (Georgescu et al., 2006; Nakamura et al., 2014; Plaschke et al., 2014); (ii) otherwise known fields, e.g., when the

spacecraft is in a diamagnetic cavity (Goetz et al., 2016a, b); or (iii) field estimates from a field model such as the International Geomagnetic Reference Field (e.g., Thébault et al., 2015).

This paper deals with option (1). It shall address the following question: How much solar wind data are needed to obtain all three components of the offset vector with a certain accuracy? It shall be assumed that the magnetometer is otherwise perfectly calibrated ($\mathbf{C}$ accurately determined), that the offset components are not drifting (i.e., non-drifting instrument and

non-varying spacecraft fields at the magnetometer sensor position), and that the magnetometer is mounted on a non-spinning spacecraft. The latter assumption means that the spacecraft spin cannot be used to support the determination of the spin plane offset components.

## 2   Data, Methods, and Results

To answer the question posed in the introduction, well-calibrated magnetometer measurements in the solar wind are needed.

In this paper, measurements from the National Aeronautics and Space Administration (NASA) Magnetospheric Multiscale (MMS) mission are used (Burch et al., 2016). The mission consists of four spin-stabilized spacecraft, launched on 13 March 2015 into highly elliptical and roughly equatorial orbits. The goal of the mission is to explore the small-scale physics of magnetic reconnection. To achieve this goal, the spacecraft are required to fly in close configuration (spacecraft separations down to a few km) in regions where reconnection is likely to take place, at the dayside magnetopause and in the geomagnetic

tail. Due to the small spacecraft separations, high cadence measurements and most accurate calibrations of all instruments are key. Otherwise, differences between spacecraft cannot be resolved. Consequently, the MMS spacecraft carry most advanced instruments to measure particle distribution functions (Pollock et al., 2016; Young et al., 2016; Mauk et al., 2016; Blake et al., 2016; Torkar et al., 2016) as well as electric and magnetic fields (Torbert et al., 2016a; Russell et al., 2016; Le Contel et al., 2016; Ergun et al., 2016; Lindqvist et al., 2016; Torbert et al., 2016b).

Here, only measurements by the MMS fluxgate magnetometers (FGM) are used. Each spacecraft is equipped with two magnetometers, an analog fluxgate and a digital fluxgate magnetometer (AFG and DFG), mounted at the ends of two separate 5 m long booms (Torbert et al., 2016a; Russell et al., 2016). The instruments and, in particular, the offsets pertaining to AFG and DFG on all spacecraft are arguably optimally calibrated: As the MMS spacecraft are spinning, the spin plane offsets can be





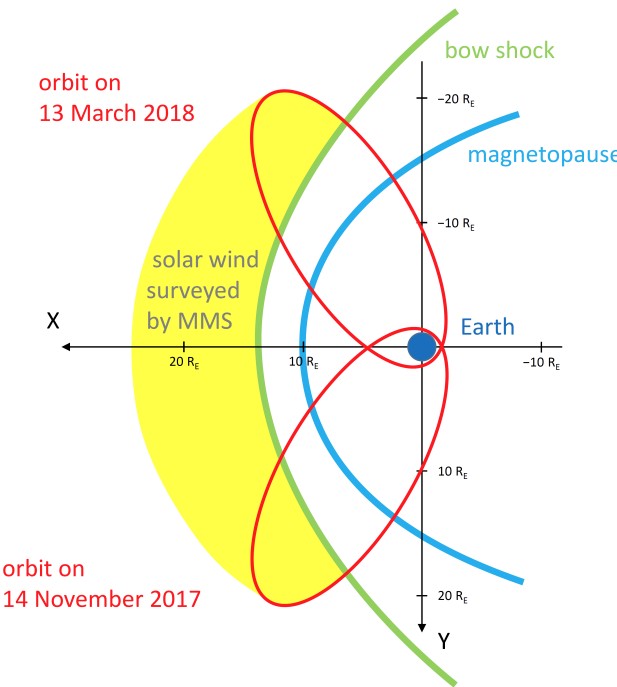

**Figure 1.** Sketch of the MMS extended mission phase 3B orbits between 14 November 2017 and 13 March 2018. Locations where MMS surveyed the solar wind in that phase are displayed in yellow. The Sun is to the left.

and are dynamically adjusted in low fields (e.g., in the solar wind). Furthermore, the spin axis offsets are updated regularly by comparison with MMS EDI measurements (Plaschke et al., 2014; Torbert et al., 2016b). Due to the small spacecraft separations, inter-spacecraft and inter-instrument (AFG versus DFG) comparisons allow for further fine tuning in the spin axis offsets. Altogether, it is not overstated that the goal of the MMS magnetic field measurements - to achieve absolute accuracies better than $0.1\,\mathrm{nT}$ in low field regions - is practically always fulfilled. Hence, any additional offsets determined from these data should ideally vanish. Deviations from 0 are, hence, indicative of the accuracy of the offset determinations.

Long-duration solar wind measurements are obtained by MMS during the extended mission phase, when the MMS orbit apogees, at distances of $25\,R_{\mathrm{E}}$ (Earth radii) from the Earth's center, are located dayside of the terminator. The perigee distances in this phase are just over $1{,}000\,\mathrm{km}$ above ground (see Figure 1). Fully calibrated MMS 1 FGM survey mode data (AFG measurements) from the dayside extended mission phase 3B (14 November 2017 to 13 March 2018, i.e., 119 days) are used in despun major principal axis (DMPA) coordinates. In this coordinate system, the major principal axis of inertia (i.e., the spin axis) points in the $z$-direction and the spacecraft-Sun vector lies in the $x$-$z$ plane. The data are available in $16\,\mathrm{Hz}$ resolution. Alfvénic fluctuations in the solar wind are of significantly lower frequency. Therefore, the data are resampled to $1\,\mathrm{Hz}$ to reduce computational efforts.





The data are subdivided into 171,360 non-overlapping one-minute intervals. Solar wind intervals are easily identified by $|\boldsymbol{B}| < 10\,\text{nT}$: $N_\text{tot} = 116{,}914$ intervals fulfill this criterion. For each of these intervals, 3D offset vector estimates $\boldsymbol{O}$ are determined by minimization of the standard deviation of $|\boldsymbol{B} - \boldsymbol{O}|$. Offset components $O_x$, $O_y$, and $O_z$ are required to be within $\pm 10\,\text{nT}$ around 0. This is fulfilled in $N_O = 68{,}324$ cases. This offset criterion selects intervals containing Alfvénic fluctuations. In these cases, a minimum of the standard deviation of $|\boldsymbol{B} - \boldsymbol{O}|$ can be found for small offset corrections $\boldsymbol{O}$. On the contrary, if there are compressional fluctuations, maximizing the offset component in the minimum variance direction will yield smallest standard deviations of $|\boldsymbol{B} - \boldsymbol{O}|$. But then, at least one component of $\boldsymbol{O}$ is likely to be found outside of $\pm 10\,\text{nT}$ or no convergence is found on any $\boldsymbol{O}$ vector altogether.

For an offset component estimate to be meaningful (e.g., $O_x$), the magnetic field in that component (e.g., $B_x$) should be fluctuating during the one-minute interval of interest. Hence, offset components pertaining to intervals are selected, where the standard deviations $\sigma$ of the respective component of $\boldsymbol{B}$ is larger than a certain threshold $\sigma_\text{c}$. The numbers $N$ of intervals selected are shown in Figure 2a. $N(\sigma_\text{c} = 0)$ is obviously $N_O = 68{,}324$ for all $x$, $y$, and $z$ components (shown in blue, green, and red, respectively); $N$ decreases if higher threshold values $\sigma_\text{c}$ are used. This decrease is not exactly the same for all components. Apparently, magnetic field fluctuations in $B_x$ are slightly weaker than in the other components, so that $N_x < N_y < N_z$ for any given $\sigma_\text{c} \neq 0$.

The numbers $N$ are fractions of all one-minute intervals of solar wind data $N_\text{tot}$, where $|\boldsymbol{B}| < 10\,\text{nT}$. Furthermore, using magnetic field data from the NASA's OMNI high resolution data set (King and Papitashvili, 2005) for the same period of time (14 November 2017 to 13 March 2018), it is possible to obtain the fraction of solar wind with $|\boldsymbol{B}| < 10\,\text{nT}$: it is 88.4%. Therewith, it is possible to compute the amount of solar wind measurements ($T$ in minutes) required to obtain one interval featuring an offset estimate within $\pm 10\,\text{nT}$ and $\sigma > \sigma_\text{c}$ in a magnetic field component:

$$T = \frac{N_\text{tot}}{0.884\,N_x} \tag{2}$$

This function $T(\sigma_\text{c})$ is shown in Figure 2b. If $\sigma_\text{c} = 0.5\,\text{nT}$, then obtaining one suitable offset estimate in any component requires almost 10 minutes of solar wind measurements.

Note that the OMNI solar wind data set from NASA's Goddard Space Flight Center (GSFC) is based on measurements by different solar wind monitors (e.g., the Advanced Composition Explorer (ACE) and the Wind spacecraft). These measurements are propagated in time to represent observations at the Earth's bow shock nose. The OMNI data set pertains to and is distributed by the NASA/GSFC's Space Physics Data Facility.

The offset estimates from any particular selected interval are almost certainly not accurate, but a sample of those intervals can yield an accurate offset. From $W$ offsets pertaining to one component ($x$, $y$, or $z$) from intervals with $\sigma > \sigma_\text{c}$, a final offset $O_\text{f}$ can be computed by using the kernel density estimator (KDE) method. From the $W$ offsets (index: $i = 1 \ldots W$), a probability density function $P$ can be determined as follows (e.g., Plaschke and Narita, 2016):

$$P(\tilde{O}) = \frac{1}{\sqrt{2\pi}Wh} \sum_{i=1}^{W} \exp\left[-\frac{1}{2}\left(\frac{\tilde{O} - O_i}{h}\right)^2\right] \tag{3}$$

The parameter $h$ is a bandwidth, set to $1\,\text{nT}$. Then $O_\text{f} = \tilde{O}$ where $P$ maximizes.



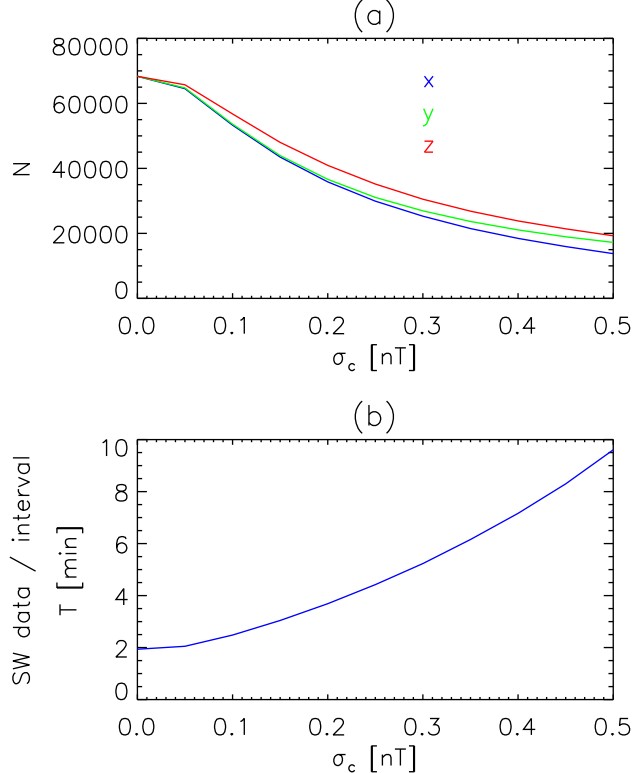

**Figure 2.** Panel a: Numbers $N$ of one-minute intervals selected when using the threshold $\sigma_c$ on component standard deviations $\sigma$ in magnetic field; $N$ pertaining to $x$, $y$, and $z$ components shown in blue, green, and red, respectively. Panel b: Minutes of solar wind data $T$ required to obtain 1 selected interval as a function of $\sigma_c$, derived from $x$-component $N(\sigma_c)$ values.

For $\sigma_c = 0 \ldots 0.5\,\mathrm{nT}$, $W = 10 \ldots 10{,}000$ offsets from each component are randomly selected 1,000 times from the available $N(\sigma_c)$ samples. Hence, for each combination of $\sigma_c$ and $W$, 1,000 estimates (index $j = 1 \ldots 1{,}000$) of $O_{xfj}$, $O_{yfj}$, and $O_{zfj}$ are computed. The maximum offset (deviations from 0) are stored:

$$O_{\max}(\sigma_c, W) = \max\left(|O_{afj}| : a \in \{x, y, z\}, j = 1 \ldots 1{,}000\right) \tag{4}$$

5   This is the upper limit estimate of the uncertainty in offset determination in any component. The values of $O_{\max}$ are displayed in Figure 3; they top out at $2\,\mathrm{nT}$. The minimum $O_{\max}$ found is $0.12\,\mathrm{nT}$. Unsurprisingly, larger sample sizes $W$ of offset estimates yield more accurate offsets, i.e., lower uncertainties $O_{\max}$. Furthermore, for constant $W$, $O_{\max}$ decreases if $\sigma_c$ is increased from 0 to approximately $0.15\,\mathrm{nT}$.

The more offset estimates $W$ from one-minutes intervals are used, the more solar wind measurements are required to obtain
10   them in the first place. Multiplying $T(\sigma_c)$ by $W$ yields that minimum solar wind measurement time. It is displayed in Figure 4. The minimum $TW$ is just over 19 minutes (for $\sigma_c = 0\,\mathrm{nT}$ and $W = 10$) and the maximum $TW$ in the figure, just over $1{,}600$ hours, pertains to $\sigma_c = 0.5\,\mathrm{nT}$ and $W = 10{,}000$.


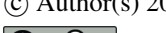

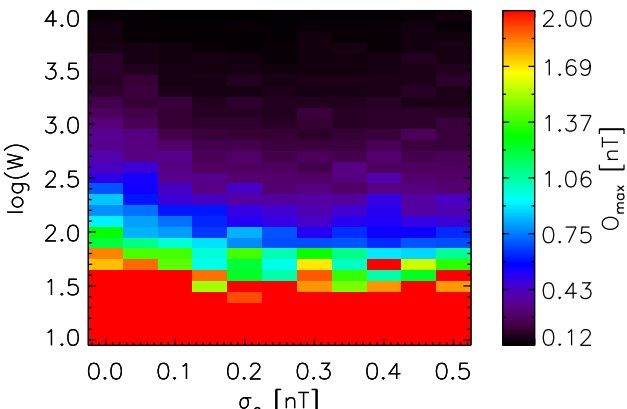

**Figure 3.** Offset uncertainties $O_{\max}$ as a function of $\sigma_c$ and $W$.

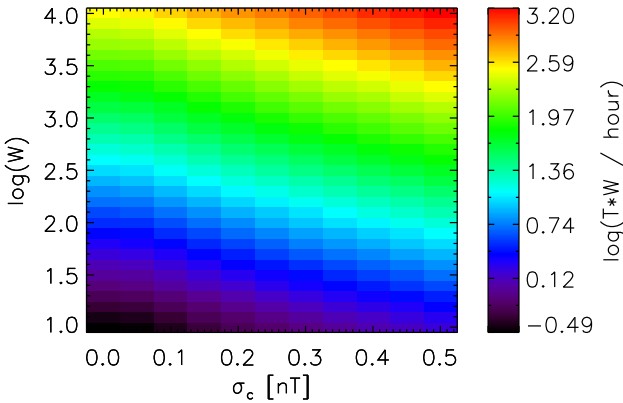

**Figure 4.** Required solar wind measurement time $T(\sigma_c)W$ to obtain $W$ offset estimates from intervals with $\sigma > \sigma_c$, from which final offsets may be computed in all components.

From the data underlying both figures it is possible to find lowest required solar wind measurement times $TW$ for given $O_{\max} \leq O_{\max,c}$, as follows: Find all combinations of $\sigma_c$ and $W$ in Figure 3 fulfilling $O_{\max} \leq O_{\max,c}$. From those combinations, identify the one associated with the lowest time $TW$ in Figure 4. The identified parameters $\sigma_c$, $W$, as well as the minimum times $TW$ are shown for different limits $O_{\max,c}$ in Table 1. Here, $O_{\max,c}$ is a threshold value for the uncertainty in the offset determination.

## 3 Discussion and Conclusions

As can be seen in Table 1, $\sigma_c = 0.15\,\mathrm{nT}$ seems to be an optimal choice. This is already visible in Figure 3, where $O_{\max}$ values appear to stay relatively constant for $\sigma_c \geq 0.15\,\mathrm{nT}$, but are noticeably larger for lower threshold values. For $\sigma_c = 0.15\,\mathrm{nT}$, $O_{\max}$ are shown as a function of $W$ or, alternatively, $TW$ in Figure 5.



**Table 1.** Optimal parameters $\sigma_{\mathrm{c}}$ and $W$ as well as minimum solar wind measurement times $TW$ to achieve offset uncertainties $\leq O_{\mathrm{max,c}}$.

| $O_{\mathrm{max,c}}$ [nT] | $\sigma_{\mathrm{c}}$ [nT] | $W$ | $TW$ [h] |
|---|---|---|---|
| 0.12 | 0.10 | 10,000 | 413.5 |
| 0.15 | 0.15 | 3,162 | 160.3 |
| 0.20 | 0.15 | 1,000 | 50.7 |
| 0.25 | 0.15 | 630 | 31.9 |
| 0.30 | 0.15 | 398 | 20.2 |
| 0.40 | 0.20 | 199 | 12.2 |
| 0.50 | 0.10 | 199 | 8.2 |
| 0.75 | 0.15 | 100 | 5.1 |
| 1.00 | 0.15 | 50 | 2.5 |
| 1.50 | 0.10 | 50 | 2.1 |

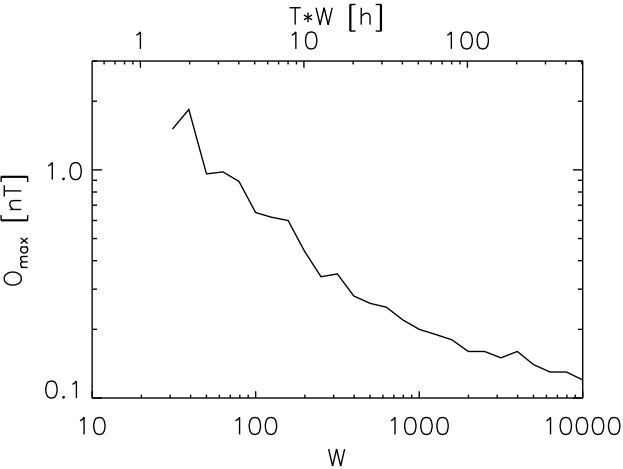

**Figure 5.** Offset uncertainties $O_{\mathrm{max}}$ as a function of $W$ or $TW$ for fixed $\sigma_{\mathrm{c}} = 0.15\,\mathrm{nT}$.

With Table 1 and Figure 5 it is possible to answer the question posed in the introduction section: How much solar wind data are needed to obtain all three components of the offset vector with a certain accuracy?

Offset determinations with uncertainties better than $O_{\mathrm{max,c}} = 0.2\,\mathrm{nT}$ are possible based on just over 2 days (50.7 hours) of solar wind measurements. If only 20.2 hours of data are available, then offsets may be determined more accurately than $O_{\mathrm{max,c}} = 0.3\,\mathrm{nT}$. Ensuring uncertainties to be significantly lower than $O_{\mathrm{max,c}} = 0.2\,\mathrm{nT}$, however, may require prohibitively long solar wind measurement intervals of several hundred hours, over which the instrument offsets and spacecraft fields at the magnetometer sensor need to stay constant to within $O_{\mathrm{max,c}}$. Otherwise, intrinsic offset drifts and field variations over time would limit the attainable accuracy, irrespective of the amount of solar wind data used.





Although MMS 1 data are used as high-quality standard to ascertain the accuracy of the offset determination with the outlined method, the results shown in Table 1 or Figure 5 are not MMS specific. They should be applicable to any magnetometer/spacecraft configuration, as long as spacecraft-generated magnetic field variations within one-minute intervals are of significantly lower amplitude in the magnetometer data than the natural magnetic field variations of solar wind Alfvénic fluc-
5   tuations. Those spacecraft-generated field variations may be sufficiently reduced by making use of double sensor gradiometer measurements.

*Data availability.*  MMS FGM level 2 survey data are publicly available at https://lasp.colorado.edu/mms/sdc/public. OMNI high resolution solar wind data are publicly available at https://omniweb.gsfc.nasa.gov.

*Competing interests.*  The author declares that he has no conflict of interest.

10  *Acknowledgements.*  The dedication and expertise of the Magnetopheric MultiScale (MMS) development and operations teams are greatly appreciated. The use of level 2 survey Flux-Gate Magnetometer (FGM) data is acknowledged. The Austrian part of the development, operation, and calibration of the Digital Flux-Gate (DFG) was financially supported by rolling grant of the Austrian Academy of Sciences and the Austrian Space Applications Programme of the Österreichische Forschungsförderungsgesellschaft (FFG) with the contract FFG/ASAP-844377. The use of NASA/GSFC's Space Physics Data Facility's OMNIWeb service and OMNI data is acknowledged.





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
