# Peer review of "How much solar wind data are sufficient for accurate fluxgate magnetometer offset determinations?"

_Geoscientific Instrumentation, Methods and Data Systems, 2019_

## Referee Comment (RC1) · Anonymous Referee #1 · 12 Apr 2019

General Comments

——————-

This paper represents an interesting new statistical approach to determining how much data is required to calibrate a fluxgate magnetometer in the solar wind, leveraging the highly accurate data set available from MMS. The paper is well presented, and the methods used to create the results are appropriate and accurately described. Some additional details may help the reader understand both the instrumental and geophysical implications of the results. I question whether the conclusions of this paper might be altered slightly after a more careful assessment of the effect of the accuracy of the

MMS data on the results. I hope this might be accomplished simply, by showing and discussing how the spin axis components and the spin plane components contribute individually to what is currently presented as "the upper limit estimate of the uncertainty in offset determination in any component".

Specific Comments
* * *
Page 3, lines 1-6:

I read the accuracy goal for MMS magnetic field measurements as a *relative* accuracy goal among the 4 spacecraft that comprise the tetrahedron (Torbert et al, 2016a), rather than an *absolute* accuracy goal, as stated here. In particular, AFG/DFG comparison ensures only relative accuracy, while the EDI comparison only claims to provide accurate FGM calibrations to 0.1 nT at best (Plaschke et al., 2014).

Furthermore, it is known that the MMS calibration accounts for temperature fluctuations on the spin-plane sensors on the order of +/- 0.2 nT, even on orbits with minimal temperature variations. It is likely that similar fluctuations exist on the spin axis component, although these offset variations are only corrected in the spin plane. Note that the temperature-dependent variations are effectively DC offsets at the 1-minute time scale (Bromund,et al., 2016). So I hesitate to agree that "additional offsets derived from these data should ideally vanish", except perhaps in the spin plane (see discusion of Page 4, line 1).

Page 3, line 12:

It is also important to note that the DMPA coordinate system is "Near GSE". Specifically, the spacecraft-sun vector is nearly aligned with the x-direction.

Page 4, line 1:

Regarding the choice of one-minute intervals, it is important to point out that corre-

sponds to ∼3 complete spin periods of MMS (Tooley et al., 2015). This is significant, because the known, ∼0.1 nT inaccuracies in the spin-plane components are manifest in DMPA coordinates as oscillations at the spin period in Bx and By, and thus will average out over each 1-minute interval. The level of attenuation would be a factor of 10 or more (depending on the exact spin period). At the same time, any DC offset on the spin axis would remain unattenuated. Thus, I would expect that the offsets derived on 1-minute intervals from the MMS data should ideally vanish to the level of 0.01 nT or less in the spin plane, while they might be as large as 0.2 nT on the spin axis.

Page 4, lines 2-3

"vector estimates O are determined by minimization of the standard deviation of |B−O|". It would be helpful to mention that this is the Davis-Smith method, given that a few distinct methods were cited on Page 2, line 3 (Belcher, 1973; Hedgecock, 1975; Leinweber et al., 2008).

Page 4, lines 14-15

"Apparently, magnetic field fluctuations in Bx are slightly weaker than in the other components, so that Nx < Ny < Nz " Could this be a natural consequence of the fact that the Bx component in the DMPA system is closely aligned with the radial direction to the sun (see above note to page 3, line 12), and that the fluctuations of the solar wind are predominantly transverse to the radial (Belcher, 1973)?

Figure 3:

I would plot a vertical line at sigma_c = 0.15 nT to further emphasize that this represents the optimal choice for sigma_c, and to better illustrate the relationship between Figure 3 and Figure 5.

Page 6, lines 6-7:

It is interesting that further increase in sigma_c beyond 0.15 nT does not result in improved accuracy. The fact that this result is so close to the expected MMS FGM

accuracy makes we wonder if this result is a function of the MMS FGM accuracy itself, or if it is inherent to the solar wind calibration process that is under evaluation. Some discussion would be helpful.

Page 8, line 2-5:

The conclusion that the results presented in Table 1 or Figure 5 are not MMS specific is not fully supported by the present analysis. Even if the absolute accuracy of the MMS data is always better than 0.1 nT as stated in this paper, this is a significant fraction of the best values of O_max,c at 0.12 nT, thus the flattening of the curve in Figure 5 at larger values of W may be due to the inherent inaccuracy of the MMS data.

Furthermore, as noted above, the spin axis likely includes temperature-dependent offset variations that are not corrected, thus I would expect O_max,c might be as large as a few hundred pT due to these effects alone, when calculated on a small number of intervals, W. These fluctuations would naturally average out as W increases, resulting in a trend that is similar to what we see in Figure 5, again calling to question the degree to which the results in Figure 5 are not MMS specific.

Inaccuracies due to the limitations of the solar wind calibration technique itself might tend to be larger in the spin plane, which includes the radial direction from the sun (Belcher, 1973). At the same time, inaccuracies in the MMS data would tend to manifest much more significantly in the spin axis. Thus, I believe it would be very useful to show the degree to which O_max,c is influenced by the spin axis and spin plane components separately. If the results of this additional analysis show that O_max,c is dominated by the spin plane components (Bx in particular), then I would be more confident that the results are not MMS specific.

Technical Corrections
* * *
Page 2, line 27-28 the wording "the MMS spacecraft carry most advanced instruments"

is a bit awkward and vague...

Page 2, line 30-31 minor edit: replace "Each spacecraft is equipped with two magnetometers" with "Each spacecraft is equipped with two fluxgate magnetometers"

Additional References

———————

C.R. Tooley, et al., The MMS observatory. Space Science Rev., 2015

K. R. Bromund, F. Plaschke, R. J. Strangeway, B. J. Anderson, B. G. Huang, W. Magnes, D. Fischer, R. Nakamura, H. K. Leinweber, C. T. Russell, W. Baumjohann, M. Chutter, R. B. Torbert, G. Le, J. A. Slavin, E. L. Kepko (2016) "In-Flight Calibration Methods for Temperature-Dependent Offsets in the MMS Fluxgate Magnetometers", Abstract SM21A-2455 presented at 2016 Fall Meeting, AGU, San Francisco, Calif., 12-16 Dec.

---

## Referee Comment (RC2) · Anonymous Referee #2 · 9 Aug 2019

Article Review Journal: GI Title: How much solar wind data are sufficient for accurate fluxgate magnetometer offset determinations? Author(s): Ferdinand Plaschke MS No.: gi-2019-4 MS Type: Research article

This paper is a very interesting work which introduce the accuracy of a method for in-orbit offset calibration by means of the Alfvénic fluctuations in the solar wind. The work is based on the data provided by the MMS NASA's mission and a statistical method to answer the question of how much solar wind data is sufficient for in-orbit calibration of space magnetometers offsets. This work is supported by the high resolution of the MMS instrument data and the comparison of these results with the NASA's OMNI

database. I found this work a worth it publication paper, as it is supported by a very well described method and conclusions, and a well selected set of references.

I would like to highlight only one comment on an aspect of the reliability criteria of the MMS data: Although the spacecrafts are not three axis stabilized, the MMS satelites spin and the axis are introduced in the work (page 3, lines 11-12): "In this coordinate system, the major principal axis of inertia (i.e., the spin axis) points in the z-direction and the spacecraft-Sun vector lies in the x-z plane". In page 4 it is mentioned that "magnetic field fluctuations in Bx are slightly weaker than in the other components, so that Nx < Ny < Nz". It would be useful for the reader to have a more detailed explanation of how the axis of the spacecraft are aligned with the spacecraft-sun direction in the orbit, as intuitively the magnetic field fluctuations in the z–x and y-z planes should be the shame. In page 3 it is mentioned that ", any additional offsets determined from these data should ideally vanish". I found this a compromised comment as it is well known the high dependence of the magnetic response of the fluxgate magnetometer with temperature. Russell et al. (Space Sci Rev (2016) 199: 189. https://doi.org/10.1007/s11214-014-0057-3) introduce the offset drift with sensor temperature as < 10pT/0C. Variations of tens of Celsius degrees (easily reachable in orbit) could lead, without the proper thermal stabilization, to a source of error bigger than the Alfvénic fluctuations in the worst case. Other revisions: Page 4, line 28: "The offset estimates estimated from any particular selected interval are almost certainly not accurate, but a sample of those intervals can yield an accurate offset" Page 5, line 9: "The more offset estimates W from one-minutes intervals are used"

Please also note the supplement to this comment:
https://www.geosci-instrum-method-data-syst-discuss.net/gi-2019-4/gi-2019-4-RC2-supplement.pdf

---

## Author Comment (AC1) · 4 Sep 2019

**Response to the reviewers' comments**

First of all, I would like to thank both reviewers for their comments and questions, which have helped me to significantly improve the manuscript. The reviewers' comments are given in bold face below, and the responses are given in blue normal type. Changes to the manuscript are shown in green. All page and line numbers refer to the original manuscript.

**Referee #1**

**General Comments**

**This paper represents an interesting new statistical approach to determining how much data is required to calibrate a fluxgate magnetometer in the solar wind, leveraging the highly accurate data set available from MMS. The paper is well presented, and the methods used to create the results are appropriate and accurately described. Some additional details may help the reader understand both the instrumental and geophysical implications of the results. I question whether the conclusions of this paper might be altered slightly after a more careful assessment of the effect of the accuracy of the MMS data on the results. I hope this might be accomplished simply, by showing and discussing how the spin axis components and the spin plane components contribute individually to what is currently presented as "the upper limit estimate of the uncertainty in offset determination in any component".**

**Specific Comments**

**Page 3, lines 1-6:**

**I read the accuracy goal for MMS magnetic field measurements as a \*relative\* accuracy goal among the 4 spacecraft that comprise the tetrahedron (Torbert et al, 2016a), rather than an \*absolute\* accuracy goal, as stated here. In particular, AFG/DFG comparison ensures only relative accuracy, while the EDI comparison only claims to provide accurate FGM calibrations to 0.1 nT at best (Plaschke et al., 2014).**

**Furthermore, it is known that the MMS calibration accounts for temperature fluctuations on the spin-plane sensors on the order of +/- 0.2 nT, even on orbits with minimal temperature variations. It is likely that similar fluctuations exist on the spin axis component, although these offset variations are only corrected in the spin plane. Note that the temperature-dependent variations are effectively DC offsets at the 1-minute time scale (Bromund, et al., 2016). So I hesitate to agree that "additional offsets derived from these data should ideally vanish", except perhaps in the spin plane (see discussion of Page 4, line 1).**

I agree with the reviewer. There is a clear difference between the attainable accuracies in the spin plane (DMPA x and y) and the spin axis (DMPA z) components.

In the spin plane, the offset determination is aided by the spacecraft spin, as any offset error leads to spin tone in these components. Hence, in low and quiet fields – like in the solar wind – the spin plane offsets can be continuously tracked and linked to temperature variations. This knowledge is then applied to dynamically correct the temperature-dependent spin plane offsets, leading to offset uncertainties on the order of some 10 pT at any given time in the region of interest.

The uncertainties in the spin axis component are larger by an order of magnitude. The offsets in this component are derived by comparison with EDI measurements, performed at certain times at the edges of the regions of interest when EDI was operated in E-field-mode. At these points in time, offsets may be obtained with accuracies on the order of 0.1 nT. These offsets cannot be determined for other times with EDI, and hence, there is no temperature-dependent offset model that can be used to dynamically correct the spin axis offset over entire regions of interest. As pointed out by the reviewer, the offsets may drift due to temperature variations by as much as 0.2 nT; this is observed in the spin plane and can reasonably be assumed to occur also with respect to the spin axis component, where it cannot be corrected.

Hence, the actual absolute uncertainty of the magnetic field measurements during any 1 minute interval may realistically be on the order of some 10 pT in the spin plane components and on the order of 0.2 to 0.3 nT in the spin axis component. Consequently, I agree with the reviewer that the statement "additional offsets derived from these data should ideally vanish" is too optimistic.

The above discussion is now reflected in the last/first paragraph of page 2/3:

…The instruments and, in particular, the offsets pertaining to AFG and DFG on all spacecraft are very well calibrated: As the MMS spacecraft are spinning, the spin plane offsets can be and are dynamically adjusted in low fields, e.g., in the solar wind (Bromund et al., 2016; Plaschke et al., 2019). Thereby, temperature dependent offset variations on the order of 0.2nT are corrected. As a result, the absolute uncertainties of the magnetic field measurements in the spin plane components can realistically be assumed to be on the order of some 10pT. The spin axis offsets are updated once per orbit by comparison with MMS EDI measurements (Plaschke et al., 2014; Torbert et al., 2016b). The accuracy of this offset determination is on the order of 0.1nT. Unfortunately, the comparison of EDI and FGM measurements cannot be performed continuously. Hence, the spin axis offsets cannot be dynamically adjusted. As they are likely to drift by 0.2nT, just as the spin plane offsets do, the absolute uncertainties pertaining to the spin axis components can realistically be assumed to be on the same order of 0.2 to maximally 0.3nT. Due to the small spacecraft separations, inter-spacecraft and inter-instrument (AFG versus DFG) comparisons allow for further fine tuning in the spin axis offsets. Any additional spin plane or spin axis offsets determined from these data should ideally vanish within the uncertainty limits (spin plane: some 10pT; spin axis: 0.2 to 0.3nT). Deviations from 0 above those uncertainty levels are, hence, indicative of the accuracy of the offset determinations.

Following the comment on page 8 line 2-5, I now treat the spin plane (x and y) and the spin axis (z) components separately. This is reflected in changes to the paragraph starting on page 5 line 1:

The maximum offsets (deviations from 0) pertaining to the spin plane (x and y) and spin axis (z) components are stored separately:

$$O_{max,xy}(\sigma_c, W) = \max(|O_{afj}| : a \in \{x,y\}, j = 1...1,000) \quad (4)$$

$$O_{max,z}(\sigma_c, W) = \max(|O_{zfj}| : j = 1...1,000) \quad (5)$$

These are the upper limit estimates of the uncertainty in offset determination in the spin plane and spin axis components. The values of $O_{max,xy}$ and $O_{max,z}$ are displayed in Figure 3; they top out at 2nT. The minima of $O_{max,xy}$ and $O_{max,z}$ are 0.05 and 0.12nT, respectively. Unsurprisingly, larger sample sizes W of offset estimates yield more accurate offsets, i.e., lower uncertainties $O_{max}$. Furthermore, for constant W, both $O_{max,xy}$ and $O_{max,z}$ decrease if $\sigma_c$ is increased. No improvement in $O_{max,xy}$ is apparent, however, for $\sigma_c > 0.15$nT.

Figure 3 now has two panels, showing $O_{max,xy}$ and $O_{max,z}$ separately as functions of W and $\sigma_c$. The values shown in Table 1 are now obtained from $O_{max,xy}$, i.e., Figure 2a. Hence, a few minor

changes had also to be made to the last paragraph of section 2 on page 6. Values in Table 1 have not changed for O_max,c >= 0.3 nT, but T*W values are significantly lower now for O_max,c < 0.2 nT.

It turns out that offsets with uncertainties above 0.3 nT are more easily found for the z component than for x and y, whereas under 0.2 nT, the discussed calibration differences of z with respect to x and y become an issue, as anticipated by the reviewer (see also comment on page 8, line 2-5). Consequently, I have updated Figure 5 and also the text starting at line 1 of page 7 as follows:

Figure 5 shows that the determination of DMPA z-component offsets with a certain accuracy (> 0.3nT) requires less solar wind data than the determination of x or y-component offsets. The reason might be the use of Alfvénic, i.e., transverse fluctuations of the solar wind magnetic field. As that field is typically lying in the ecliptic x-y-plane (Parker spiral interplanetary magnetic field), the transverse fluctuations will be most apparent in the normal z-component (Belcher, 1973). Accuracies better than 0.2 nT are, however, easier to achieve for the spin plane x and y components, due to the 0.2 to 0.3 nT uncertainty and variability in the MMS spin axis offsets. …

Values of O_max,xy ≥ 0.1nT should be unaffected by x and y-component calibration uncertainties of some 10pT. Hence, with the help of Table 1 and Figure 5 (blue line), both showing O_max,xy results, it is possible to give an answer to the question posed in the introduction section: How much solar wind data are needed to obtain all three components of the offset vector with a certain accuracy?

For the aim of this study, it makes sense to use only the x and y spin plane offset results, shown in Figure 5 (blue) and in Table 1. Minor modifications to the rest of the conclusions section and to the abstract were also necessary as some of the values of Table 1 have changed.

**Page 3, line 12:**

**It is also important to note that the DMPA coordinate system is "Near GSE". Specifically, the spacecraft-sun vector is nearly aligned with the x-direction.**

I agree that this is an important point. This new sentence has been inserted on page 3 line 12:

The DMPA system is closely aligned with the geocentric solar ecliptic (GSE) system, as the spin axis (z) points essentially normal to the ecliptic and the spacecraft-Sun vector almost coincides with the DMPA x-axis.

**Page 4, line 1:**

**Regarding the choice of one-minute intervals, it is important to point out that corresponds to ~3 complete spin periods of MMS (Tooley et al., 2015). This is significant, because the known, ~0.1 nT inaccuracies in the spin-plane components are manifest in DMPA coordinates as oscillations at the spin period in Bx and By, and thus will average out over each 1-minute interval. The level of attenuation would be a factor of 10 or more (depending on the exact spin period). At the same time, any DC offset on the spin axis would remain unattenuated. Thus, I would expect that the offsets derived on 1-minute intervals from the MMS data should ideally vanish to the level of 0.01 nT or less in the spin plane, while they might be as large as 0.2 nT on the spin axis.**

I mention now in the manuscript (on page 4, line 1) that 1 minute is a multiple of the spacecraft spin period. However, I do not think that it helps to attenuate any residual offset effects, as I do not average magnetic field data over one minute.

Note that one minute is a multiple of the spacecraft spin period of 20 seconds (Tooley et al., 2016).

**Page 4, lines 2-3**

**"vector estimates O are determined by minimization of the standard deviation of |B−O|". It would be helpful to mention that this is the Davis-Smith method, given that a few distinct methods were cited on Page 2, line 3 (Belcher, 1973; Hedgecock, 1975; Leinweber et al., 2008).**

I agree with the reviewer. It is now mentioned on page 4 line 3.

**Page 4, lines 14-15**

**"Apparently, magnetic field fluctuations in Bx are slightly weaker than in the other components, so that Nx < Ny < Nz " Could this be a natural consequence of the fact that the Bx component in the DMPA system is closely aligned with the radial direction to the sun (see above note to page 3, line 12), and that the fluctuations of the solar wind are predominantly transverse to the radial (Belcher, 1973)?**

Yes, that is what I also suspect. Consequently, I have added the following sentence to page 4 line 15:

Note that the DMPA x-component corresponds with the radial direction to the Sun, which has previously been reported to feature lower levels of fluctuations (Belcher, 1973).

**Figure 3:**

**I would plot a vertical line at sigma_c = 0.15 nT to further emphasize that this represents the optimal choice for sigma_c, and to better illustrate the relationship between Figure 3 and Figure 5.**

I agree with the reviewer. Vertical white lines mark sigma_c = 0.15 nT in Figures 3a and b. Additional text on page 5 line 8 and at the end of page 6 refers to these lines.

**Page 6, lines 6-7:**

**It is interesting that further increase in sigma_c beyond 0.15 nT does not result in improved accuracy. The fact that this result is so close to the expected MMS FGM accuracy makes we wonder if this result is a function of the MMS FGM accuracy itself, or if it is inherent to the solar wind calibration process that is under evaluation. Some discussion would be helpful.**

Interestingly, the sigma_c limit does apply to O_max,xy but not really to O_max,z, although it is the z-component that has a similar uncertainty (see new Figure 3). Hence, I would hypothesize that this limit is not MMS specific but may have some relation to the solar wind fluctuations. I have added the following text to the beginning of page 7:

Note, that the $\sigma_c = 0.15$nT limit does not seem to be related to this uncertainty, as it is more visible in the O_max,xy than in the O_max,z results, presented in Figures 3a and b, respectively. Instead, it may be hypothesized that the optimal choice of $\sigma_c$ is rather related to typical fluctuation amplitudes of the solar wind magnetic field.

**Page 8, line 2-5:**

**The conclusion that the results presented in Table 1 or Figure 5 are not MMS specific is not fully supported by the present analysis. Even if the absolute accuracy of the MMS data is always better than 0.1 nT as stated in this paper, this is a significant fraction of the best values of $O_{max,c}$ at 0.12 nT, thus the flattening of the curve in Figure 5 at larger values of W may be due to the inherent inaccuracy of the MMS data.**

**Furthermore, as noted above, the spin axis likely includes temperature-dependent offset variations that are not corrected, thus I would expect $O_{max,c}$ might be as large as a few hundred pT due to these effects alone, when calculated on a small number of intervals, W. These fluctuations would naturally average out as W increases, resulting in a trend that is similar to what we see in Figure 5, again calling to question the degree to which the results in Figure 5 are not MMS specific.**

**Inaccuracies due to the limitations of the solar wind calibration technique itself might tend to be larger in the spin plane, which includes the radial direction from the sun (Belcher, 1973). At the same time, inaccuracies in the MMS data would tend to manifest much more significantly in the spin axis. Thus, I believe it would be very useful to show the degree to which $O_{max,c}$ is influenced by the spin axis and spin plane components separately. If the results of this additional analysis show that $O_{max,c}$ is dominated by the spin plane components (Bx in particular), then I would be more confident that the results are not MMS specific.**

I completely agree with the reviewer. Results are now shown separately for the spin plane and spin axis components. As anticipated by the reviewer, the limitations of the method are larger in the spin plane, whereas the MMS calibration uncertainties are more significant in the spin axis. The new Figure 5 shows that $O_{max}$ is dominated by the spin plane, but that the spin axis uncertainties led to an MMS-specific over-estimation of the required solar wind measurement times in the original manuscript. In the revised manuscript, Table 1 data are based on spin plane offsets only, yielding MMS-independent results down to accuracies of approximately 0.1 nT.

All modifications related to this comment have already been introduced above.

**Technical Corrections**

**Page 2, line 27-28 the wording "the MMS spacecraft carry most advanced instruments" is a bit awkward and vague...**

Changed.

**Page 2, line 30-31 minor edit: replace "Each spacecraft is equipped with two magnetometers" with "Each spacecraft is equipped with two fluxgate magnetometers"**

Done.

**Referee #2**

**Article Review Journal: GI Title: How much solar wind data are sufficient for accurate fluxgate magnetometer offset determinations? Author(s): Ferdinand Plaschke MS No.: gi-2019-4 MS Type: Research article**

This paper is a very interesting work which introduce the accuracy of a method for inorbit offset calibration by means of the Alfvénic fluctuations in the solar wind. The work is based on the data provided by the MMS NASA's mission and a statistical method to answer the question of how much solar wind data is sufficient for in-orbit calibration of space magnetometers offsets. This work is supported by the high resolution of the MMS instrument data and the comparison of these results with the NASA's OMNI database.

I found this work a worth it publication paper, as it is supported by a very well described method and conclusions, and a well selected set of references.

I would like to highlight only one comment on an aspect of the reliability criteria of the MMS data:

Although the spacecrafts are not three axis stabilized, the MMS satelites spin and the axis are introduced in the work (page 3, lines 11-12): "In this coordinate system, the major principal axis of inertia (i.e., the spin axis) points in the z-direction and the spacecraft-Sun vector lies in the x-z plane". In page 4 it is mentioned that "magnetic field fluctuations in Bx are slightly weaker than in the other components, so that Nx < Ny < Nz". It would be useful for the reader to have a more detailed explanation of how the axis of the spacecraft are aligned with the spacecraft-sun direction in the orbit, as intuitively the magnetic field fluctuations in the z–x and y-z planes should be the shame.

An explanation is now included in the manuscript. Please refer to my responses with respect to the comment on page 4, lines 14-15, and to the comment on page 3, line 12, by reviewer 1.

In page 3 it is mentioned that ", any additional offsets determined from these data should ideally vanish". I found this a compromised comment as it is well known the high dependence of the magnetic response of the fluxgate magnetometer with temperature. Russell et al. (Space Sci Rev (2016) 199: 189. https://doi.org/10.1007/s11214-014-0057-3) introduce the offset drift with sensor temperature as < 10pT/0C. Variations of tens of Celsius degrees (easily reachable in orbit) could lead, without the proper thermal stabilization, to a source of error bigger than the Alfvénic fluctuations in the worst case.

I totally agree with the reviewer. As discussed in relation to the first comment by reviewer 1, the offset variations due to temperature changes are on the order of 0.2 nT. These variations are identified and corrected in the spin plane, but cannot be corrected in the spin axis. Correspondingly, there are quite different uncertainty levels associated to the spin plane and spin axis components. I now include a separate analysis for both. Please see my response to the first comment by reviewer 1 for more details.

**Other revisions:**

**Page 4, line 28: "The offset**  **estimated from any particular selected interval are almost certainly not accurate, but a sample of those intervals can yield an accurate offset"**

**Page 5, line 9: "The more offset estimates W from one-minutes intervals are used"**

Actually, here I meant "estimates" instead of "estimated". "One-minutes" has been changed to "one-minute".